# Expert-Guided Cross-View Fusion with Self-Derived Lesion Proposals for Multi-View Diabetic Retinopathy Grading

## Abstract

Recent advances in multi-view fundus imaging show great promise for automated diabetic retinopathy (DR) grading. However, mainstream end-to-end CNN/Transformer pipelines rely on striding or tokenization that compresses spatial detail, causing small, low-contrast lesions (e.g., microaneurysms) to be underrepresented and creating performance ceilings. Prior efforts have mitigated this by incorporating external lesion- or vessel-level annotations into models. However, such labels are costly to acquire, break the end-to-end training, and make performance over-reliant on the annotation quality. To reduce dependence on expensive annotations, we propose an end-to-end framework that generates lesion proposals on the fly during training and inference, providing self-derived cues for grading. First, we introduce a Grade-Activated Lesion Proposal (GALP) module that derives grade-conditioned evidence maps (GEMs) from stage-wise auxiliary classifiers and selects the top-K high-evidence regions per view as lesion proposals. Second, we propose a Cross-View Lesion Expert Guided Regional Fusion (LGRF) module, which selectively activates experts for a view's lesion proposals based on contextual guidance from other views, ensuring that only the most relevant feature extractors contribute to fusion. Experimental results on two multiview DR datasets show that our method matches or surpasses strong baselines without external annotations, confirming that self-generated proposals can substantially reduce annotation needs.

## 1 Introduction

DR is a microvascular complication of diabetes characterized by progressive retinal damage and is a leading cause of vision impairment and blindness. Early stages are often asymptomatic; as the disease advances, patients may experience blurred or distorted vision and scotomas (Yu et al., 2024). Without timely intervention, DR can progress to vitreous hemorrhage, tractional retinal detachment, and irreversible blindness. Consequently, population-level fundus screening, particularly in primary and community settings, is essential (Zhang et al., 2024). Yet the global supply of retina specialists is insufficient to meet the rising screening demand across both high- and low-resource regions. This mismatch has motivated intensive research into automated DR grading from fundus photographs, with deep learning emerging as a prominent approach (Lin et al., 2025b).

Research on DR grading from fundus images has generally progressed through three stages. Stage I: single-view grading. Early work ingests a single fundus photograph and predicts a five-point grade (0–4: normal, mild, moderate, severe, and proliferative DR) (Dai et al., 2021). In this setting, models learn lesion patterns from one view only, which limits their ability to capture the full retinal status (Liu et al., 2025; Zou et al., 2025). Stage II: multi-view grading. To address coverage gaps, recent studies leverage multiple views and design end-to-end fusion strategies that aggregate within-view lesion evidence and learn cross-view relationships, yielding notable gains over single-view baselines Luo et al. (2021). Stage III: revisiting end-to-end limitations. Despite progress, many end-to-end CNN/Transformer pipelines rely on downsampling or tokenization that compresses spatial detail; subtle, low-contrast lesions (or vessels) may receive insufficient attention (Luo et al., 2024). As illustrated in Fig. 1 (Switch = Off), this bottleneck can persist even with multi-view inputs. A complementary line of work augments grading with additional signals, thereby improving performance

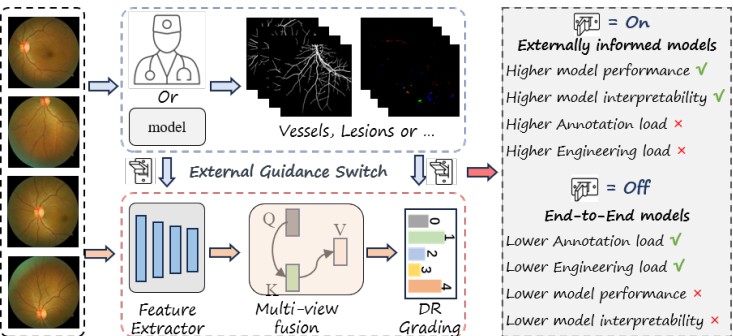

Figure 1: The comparison of the end-to-end models and the externally informed models

but at the cost of extra supervision and complexity. For instance, model proposed by Lin et al. (2025a) incorporates vessel annotations, and another work leverages clinician-annotated optic disc (OD) and macular locations (Hou et al., 2022). Such approaches (Fig. 1, Switch = On) can be effective, yet they introduce two practical challenges. First, acquisition cost and workflow burden: doctor-provided annotations are expensive and time-consuming; moreover, if inference requires those auxiliary inputs, clinicians must continue to provide them even after deployment. Second, dependency and brittleness: when auxiliary signals are produced by separate models (e.g., lesion segmenters, as in Luo et al. (2025)), grading accuracy can become tightly coupled to the upstream model's errors.

To address these limitations, we introduce a method that maintains the advantages of end-to-end multi-view learning and substantially reduces dependence on external annotations. To this end, we generate lesion-aware cues natively within the grading pipeline, targeting competitive, or superior, accuracy without external side information. Concretely, we introduce two modules: GALP and LGRF. GALP attaches stage-wise auxiliary classifiers to multi-resolution feature maps and enhances their grade-discriminative capacity via an auxiliary classification loss. From the auxiliary heads, we derive GEMs by estimating the importance of subregions with respect to the predicted grade. Since the grade evidence in DR is predominantly localized to lesions, selecting Top-$K$ peaks within these maps yields lesion proposals, whcih is the spatial regions most predictive of the grade. GALP both strengthens supervision of intermediate representations and provides proposals that act as surrogates for external cues. LGRF uses cross-view lesion proposals to guide information fusion. For each view's lesion proposals, an expert pool performs proposal-aware feature extraction; cross-view context gates which experts are activated, encouraging the current view to prioritize regions corroborated by other views. A Top$K$–weighted cross-view attention module then fuses the selected expert outputs with the current view's feature maps, achieving precise, selective integration across views. Our contributions are as follows:

(1) We propose an end-to-end DR grading framework that self-generates lesion proposals via GALP, preserving end-to-end training, strengthening intermediate representations, and recovering small, low-contrast lesions without external annotations.

(2) We introduce LGRF, a cross-view, lesion-expert–guided regional fusion module that dynamically routes experts via contextual corroboration and fuses proposals through Top-K–weighted cross-view attention, enabling precise, selective integration and superior robustness and interpretability.

(3) Comprehensive evaluations across two multi-view DR benchmarks confirm SOTA competitiveness without external supervision, showing self-derived proposals reduce annotation reliance while still elevating micro-lesion sensitivity and reliability for DR grading.

## 2 RELATED WORKS

In recent years, deep learning–based automated grading for multi-view DR has shown substantial promise (Wang et al., 2025). A major line of research exploits complementary information across standardized views by designing stronger feature extractors and cross-view fusion strategies to better

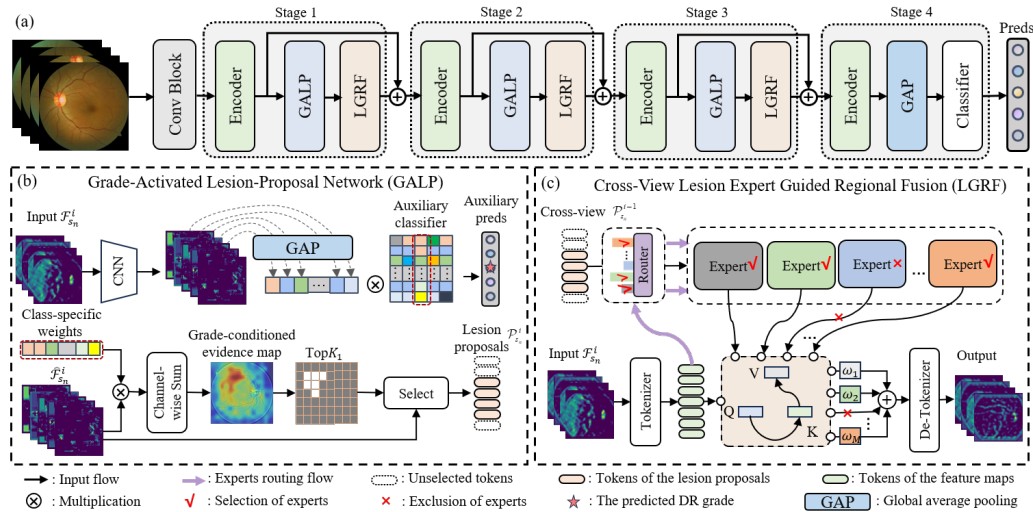

Figure 2: Overall framework. (a) Pipeline with GALP and LGRF. (b) GALP generates lesion proposals with the assistance of auxiliary classifier. (c) LGRF fuses current-view features with cross-view proposal features via gated mixture-of-experts (MoE) and Top-K-weighted cross-view attention.

capture heterogeneous lesion morphologies. To our knowledge, Luo et al. (2021) is among the earliest works to employ four-view fundus images for DR grading. Luo et al. subsequently proposed MVCINN, which hybridizes self-attention with CNNs to fuse multi-view features and improves accuracy over single-view baselines (Luo et al., 2023).

Following the era of multi-view DR recognition, a growing body of work incorporates additional signals to break performance bottlenecks. Such auxiliary cues compensate for the limited capture of fine-grained retinal structures by purely end-to-end pipelines and have delivered notable gains (Li et al., 2024; Guo et al., 2025). CVSA leverages vessel masks extracted via gaussian modeling as auxiliary inputs and introduces a cross-view lesion-alignment strategy to aggregate relevant evidence across views (Lin et al., 2025a). This yields a $2.5\%$ absolute accuracy improvement over MVCINN, a strong end-to-end state-of-the-art (SOTA) baseline. Luo et al. further proposed SMVDR (Luo et al., 2024) and LFMVDR (Luo et al., 2025), which employ lesion-annotation maps so that the model persistently attends to clinically salient regions during inference, enhancing both accuracy and interpretability. Hu et al. introduced WGLIN, a lesion-guided framework that performs wavelet-based fusion for multi-view integration (Hu et al., 2025). Distinct from methods relying on full vessel or lesion structures, Hou et al. proposed CrossFiT, which uses OD and macular coordinates to align cross-view information and also reports strong performance (Hou et al., 2022).

Despite their effectiveness, these approaches introduce practical burdens. Doctor-provided annotations are costly and time-consuming, and when auxiliary inputs are required at inference, the clinical workflow becomes heavier. Moreover, when auxiliary signals are produced by separate models (e.g., lesion segmenters), grading performance becomes tightly coupled to upstream accuracy and calibration, increasing system brittleness.

## 3 METHOD

### 3.1 OVERVIEW

The overall pipeline is shown in Fig. 2(a). Let $\mathcal{V} = \left[ \mathcal{V}^1, \mathcal{V}^2, \ldots, \mathcal{V}^N \right]$ denote the multi-view fundus images, where $\mathcal{V}^i \in \mathbb{R}^{C \times H \times W}$ is the $i$-th view and $N$ is the number of views. All views are first processed by a convolutional stem ("Conv Block") inherited from a pretrained backbone. The pipeline then proceeds through four stages, each using the backbone encoder blocks from shallow to deep to extract multi-view features. And at each stage $s_n$ ($n \in \{1, 2, 3, 4\}$), the encoded fea-

tures of view $i$, $\boldsymbol{\mathcal{F}}_{s_n}^i \in \mathbb{R}^{C_{s_n} \times H_{s_n} \times W_{s_n}}$, are then processed by the proposed GALP and LGRF. As illustrated in Fig. 2(b), GALP attaches an auxiliary classifier to stage features $\boldsymbol{\mathcal{F}}_{s_n}^i$ and optimizes an auxiliary loss so that $\boldsymbol{\mathcal{F}}_{s_n}^i$ becomes more grade-discriminative. From the auxiliary head we derive GEMs, upon which Top-$K$ region selection are applied to obtain lesion proposals $\boldsymbol{\mathcal{P}}_{s_n}^i$. As shown in Fig. 2(c), lesion proposals from the other views are processed by an expert pool whose activations are gated by the current view's features. A Top$K$–weighted cross-view attention module then fuses the selected expert outputs with the current features $\boldsymbol{\mathcal{F}}_{s_n}^i$. Finally, the stage-$s_4$ features are passed through global average pooling (GAP) and a linear classifier to produce the DR grade prediction.

## 3.2 GRADE-ACTIVATED LESION PROPOSALS (GALP)

GALP pursues two objectives: (i) enhancing $\boldsymbol{\mathcal{F}}_{s_n}^i$ with stage-wise discriminative supervision through auxiliary classification, and (ii) deriving lesion proposals that can act as surrogates for external cues.

**Auxiliary classification:** Given the encoded features $\boldsymbol{\mathcal{F}}_{s_n}^i$ from stage $s_n$ of view $i$, an auxiliary head computes logits

$$\mathbf{z}_{s_n}^i = \mathbf{W}_{s_n} \, \mathrm{GAP}\big(\mathrm{CNN}_{s_n}(\boldsymbol{\mathcal{F}}_{s_n}^i)\big), \tag{1}$$

followed by $\hat{\mathbf{y}}_{s_n}^i = \mathrm{Softmax}(\mathbf{z}_{s_n}^i)$. The auxiliary loss encourages grade-discriminative intermediate representations:

$$\boldsymbol{\mathcal{L}}_{\mathrm{aux}} = \sum_{n=1}^3 \sum_{i=1}^N \mathrm{Focal}\Big(\hat{\mathbf{y}}_{s_n}^i, \, \boldsymbol{y}\Big), \tag{2}$$

where $\boldsymbol{y}$ is the ground-truth DR grade and $\mathrm{Focal}(\cdot, \cdot)$ is the focal loss.

**lesion proposals:** Given that DR grades are predominantly determined by lesion evidence, we compute grade-conditioned regions on the feature maps in a stage-wise manner using class activation maps (CAMs) (Jiang et al., 2021). These regions are interpreted as grade-related (i.e., lesion) areas. Let $\mathbf{w}_{s_n}^{(\hat{\mathbf{y}}_{s_n}^i)} \in \mathbb{R}^{C_{s_n}}$ denote the class-specific weight vector for the predicted grade $\hat{\mathbf{y}}_{s_n}^i$ at stage $s_n$. The GEMs for view $i$ is

$$\mathbf{A}_{s_n}^i(u,v) = \mathrm{ReLU}\left(\sum_{c=1}^{C_{s_n}} \mathbf{w}_{s_n,c}^{(\hat{\mathbf{y}}_{s_n}^i)} \big(\mathrm{CNN}_{s_n}(\boldsymbol{\mathcal{F}}_{s_n}^i)\big)_{c,u,v}\right). \tag{3}$$

We normalize $\tilde{\mathbf{A}}_{s_n}^i = \big(\mathbf{A}_{s_n}^i - \min\big)/\big(\max - \min\big)$, where $\min$ and $\max$ are taken over $(u,v)$. Since $\tilde{\mathbf{A}}_{s_n}^i$ is a class-weighted sum of stage-$s_n$ feature responses contributing to the grade logit $\hat{\mathbf{y}}_{s_n}^i$, larger values mark spatial locations that increase this logit and are therefore more predictive of the grade. Accordingly, regions with higher activation in $\tilde{\mathbf{A}}_{s_n}^i$ are more likely to contain lesion evidence. Then, the spatial domain of $\tilde{\mathbf{A}}_{s_n}^i$ is partitioned into non-overlapping $q \times q$ patches (with $q$ chosen such that $q \mid H_{s_n}$ and $q \mid W_{s_n}$), yielding $P_{s_n} = (H_{s_n}/q) \times (W_{s_n}/q)$ regions $\{\Omega_{s_n}^{i,r}\}_{r=1}^{P_{s_n}}$. Therefore, lesion-likelihood score for region $r$ is defined as

$$s_{s_n}^{i,r} = \sum_{(u,v) \in \Omega_{s_n}^{i,r}} \tilde{\mathbf{A}}_{s_n}^i(u,v). \tag{4}$$

Let $\mathcal{I}_{s_n}^i = \mathrm{TopK}_{K_{1,s_n}}\big(\{s_{s_n}^{i,r}\}_{r=1}^{P_{s_n}}\big)$ be the indices of the $K_{1,s_n}$ highest-scoring regions. It is worth noting that larger values of $s_{s_n}^{i,r}$ indicate a higher likelihood that lesions reside in region $r$; thus $\mathcal{I}_{s_n}^i$ selects the most lesion-likely regions. To extract features for these regions, for each $k_{1,s_n} \in \mathcal{I}_{s_n}^i$ we compute a masked average over the encoder features:

$$\mathbf{f}_{s_n}^{i,k_{1,s_n}} = \frac{1}{|\Omega_{s_n}^{i,k_{1,s_n}}|} \sum_{(u,v) \in \Omega_{s_n}^{i,k_{1,s_n}}} \boldsymbol{\mathcal{F}}_{s_n}^i(:,u,v). \tag{5}$$

A linear projection produces $D$-dimensional tokens,

$$\mathbf{p}_{s_n}^{i,k_{1,s_n}} = \mathrm{Lr}(\mathbf{f}_{s_n}^{i,k_{1,s_n}}) \in \mathbb{R}^D, \tag{6}$$

and the proposal matrix is

$$\boldsymbol{\mathcal{P}}_{s_n}^i = \left[\mathbf{p}_{s_n}^{i,k_{1,s_n}}\right]_{k_{1,s_n} \in \mathcal{I}_{s_n}^i} \in \mathbb{R}^{K_{1,s_n} \times D}. \tag{7}$$

And the obtained $\boldsymbol{\mathcal{P}}_{s_n}^i$ can be treated as lesion proposals for downstream cross-view fusion.

Note that a uniform partition of the feature map would yield $P_{s_n}$ tokens at stage $s_n$; instead, by retaining only the Top-$K_{1,s_n}$ lesion-salient regions, we obtain $K_{1,s_n} \ll P_{s_n}$ proposal tokens that concentrate evidence on grade-relevant areas. Fusing these lesion-only tokens with cross-view features reduces distraction from non-lesion background and strengthens guidance for cross-view integration.

### 3.3 CROSS-VIEW LESION EXPERT-GUIDED REGIONAL FUSION

LGRF leverages lesion proposals produced by GALP to enable selective, proposal-aligned fusion across views. For the current view $i$ at stage $s_n$ with features $\boldsymbol{\mathcal{F}}_{s_n}^i$, we tokenize the feature map using ViT-style patching with the same patch size $q \times q$ as in proposal generation:

$$\boldsymbol{\mathcal{T}}_{s_n}^i = \text{TokN}\left(\boldsymbol{\mathcal{F}}_{s_n}^i\right) \in \mathbb{R}^{P_{s_n} \times D}, \quad P_{s_n} = \frac{H_{s_n}}{q} \cdot \frac{W_{s_n}}{q}. \tag{8}$$

Cross-view fusion is performed between the current view and its adjacent (cyclic) view $j = \begin{cases} i+1, & i < N \\ 1, & i = N \end{cases}$. We collect the Top-$K_{1,s_n}$ proposal tokens from the adjacent view, $\boldsymbol{\mathcal{P}}_{s_n}^j \in \mathbb{R}^{K_{1,s_n} \times D}$, and restrict fusion to these lesion-salient proposals to provide targeted cross-view guidance while suppressing background interference. The following subsections detail (i) Cross-view lesion proposal expert routing and (ii) Top-$K$–weighted cross-view attention.

**Cross-view lesion proposal expert routing:** To allow the current view to autonomously select which experts to activate for processing lesion proposals from the adjacent view, we gate cross-view experts conditioned on the current view's features. We first pass the current-view tokens $\boldsymbol{\mathcal{T}}_{s_n}^i$ through a routing network to determine adjacent-view expert activations. Specifically, a linear projection (denoted $\text{Router}$) maps aggregated current-view tokens to routing logits, which are then normalized via softmax to obtain routing scores:

$$\boldsymbol{\mathcal{R}s}_{s_n}^i = \text{Softmax}\left(\text{Router}\left(\text{mean}(\boldsymbol{\mathcal{T}}_{s_n}^i)\right)\right) \in \mathbb{R}^M, \tag{9}$$

where $M$ denotes the number of predefined experts in the adjacent view. Inspired by the MoE framework (Cao et al., 2023), the cross-view lesion proposals $\boldsymbol{\mathcal{P}}_{s_n}^j$ are fed into the top-$K_2$ Transformer experts $\left\{\text{Tr}_{s_n,1}^j, \text{Tr}_{s_n,2}^j, \ldots, \text{Tr}_{s_n,K_2}^j\right\}$, selected according to the $K_2$ largest entries of $\boldsymbol{\mathcal{R}s}_{s_n}^i$. The output of the $k_2$-th activated expert is

$$\boldsymbol{\mathcal{P}e}_{s_n,k_2}^j = \text{Tr}_{s_n,k_2}^j\left(\boldsymbol{\mathcal{P}}_{s_n}^j\right). \tag{10}$$

Subsequently, each extracted feature $\boldsymbol{\mathcal{P}e}_{s_n,k_2}^j$, together with its importance weight $\hat{w}_{s_n,k_2}^i$ (the $k_2$-th largest entry of $\boldsymbol{\mathcal{R}s}_{s_n}^i$) and the current-view tokens $\boldsymbol{\mathcal{T}}_{s_n}^i$, is passed to a TopK-weighted cross-view attention module to facilitate adaptive cross-view fusion. Similar to existing MoE-based methods (Xie et al., 2025), we incorporate a load-balancing loss term ($\mathcal{L}_{load}$) to encourage equitable utilization of experts. Let $B$ be the mini-batch of size, $\hat{u}_m$ be the fraction of tokens actually assigned to expert $m$ and $\boldsymbol{\mathcal{R}s}_{s_n,b,m}^i$ be the $m$-th score of the $\boldsymbol{\mathcal{R}s}_{s_n}^i$ in the $b$-th batch. The $\mathcal{L}_{load}$ is defined as

$$\mathcal{L}_{\text{load},s_n}^i = M \cdot \sum_{m=1}^{M}\left(\frac{1}{B}\sum_{b=1}^{B} \boldsymbol{\mathcal{R}s}_{s_n,b,m}^i\right) \cdot \hat{u}_m, \qquad \mathcal{L}_{\text{load}} = \sum_{n=1}^{3}\sum_{i=1}^{N} \mathcal{L}_{\text{load},s_n}^i \tag{11}$$

**Top-$K$–weighted cross-view attention:** The tokens $\boldsymbol{\mathcal{P}e}_{s_n,k_2}^j$ are projected to keys $\boldsymbol{\mathcal{K}}_{s_n,k_2}^j$ and values $\boldsymbol{\mathcal{V}}_{s_n,k_2}^j$, while the current-view tokens $\boldsymbol{\mathcal{T}}_{s_n}^i$ are projected to queries $\boldsymbol{\mathcal{Q}}_{s_n}^i$. Region-wise relationships between view $i$ and lesion regions in view $j$ are computed via

$$\boldsymbol{\mathcal{M}}_{s_n,k_2}^{ij} = \boldsymbol{\mathcal{Q}}_{s_n}^i \left(\boldsymbol{\mathcal{K}}_{s_n,k_2}^j\right)^\top \in \mathbb{R}^{P_{s_n} \times K_{1,s_n}}, \tag{12}$$

where each row of $\boldsymbol{\mathcal{Q}}_{s_n}^i$ and $\boldsymbol{\mathcal{K}}_{s_n,k_2}^j$ is $\ell_2$-normalized. With this normalization, $\boldsymbol{\mathcal{M}}_{s_n,k_2}^{ij}$ represents cosine similarities; the entry $(m, n)$ quantifies the relevance between the $m$-th region of view $i$ and the $n$-th lesion region of view $j$. Here, subscripts such as $s_n$, $i$ ($j$), and $k_2$ denote different feature stages, view indices, and expert indices, respectively, and do not refer to individual matrix entries.

For each activated expert $k_2 \in \{1, \dots, K_2\}$, attention and aggregation are

$$\mathbf{O}_{s_n,k_2}^{ij} = \mathrm{Softmax}\left(\frac{\boldsymbol{\mathcal{M}}_{s_n,k_2}^{ij}}{\sqrt{D}}\right) \boldsymbol{\mathcal{V}}_{s_n,k_2}^j \quad \in \mathbb{R}^{P_{s_n} \times D}. \tag{13}$$

Top-$K_2$ weighting by the routing scores $\hat{w}_{s_n,k_2}^i$ yields the expert-aggregated output

$$\mathbf{O}_{s_n}^{ij} = \sum_{k_2=1}^{K_2} \hat{w}_{s_n,k_2}^i \, \mathrm{FC}\left(\mathbf{O}_{s_n,k_2}^{ij}\right) \in \mathbb{R}^{P_{s_n} \times D}. \tag{14}$$

Let $\mathrm{MHA}_{\mathrm{CVA}}(\cdot)$ denote a multi-head version of the above Top-K weighted cross-view attention. Then the fused tokens for view $i$ are obtained with standard residual and layer normalization:

$$\hat{\boldsymbol{\mathcal{T}}}_{s_n}^i = \mathrm{LN}\left(\boldsymbol{\mathcal{T}}_{s_n}^i + \mathrm{MHA}_{\mathrm{CVA}}\left(\boldsymbol{\mathcal{Q}}_{s_n}^i, \{\boldsymbol{\mathcal{K}}_{s_n,k_2}^j\}, \{\boldsymbol{\mathcal{V}}_{s_n,k_2}^j\}, \{\hat{w}_{s_n,k_2}^j\}\right)\right). \tag{15}$$

Finally, tokens are reshaped back to the spatial layout to form the fused feature map:

$$\mathbf{Fu}_{s_n}^i = \mathrm{DeTok}\left(\hat{\boldsymbol{\mathcal{T}}}_{s_n}^i\right) \in \mathbb{R}^{C_{s_n} \times H_{s_n} \times W_{s_n}}. \tag{16}$$

By routing only lesion-proposal tokens and applying Top-$K$–weighted cross-view attention, the fusion focuses computation on grade-relevant regions, reducing background leakage and improving alignment fidelity.

### 3.4 DR GRADING

As shown in Fig. 2(a), the final grade is predicted from the stage-$s_4$ features by GAP, multi-view aggregation, and a linear classifier. Let $\mathrm{Concat}(\cdot)$ the channel-wise concatenation. For view $i \in \{1, \dots, N\}$, define $\mathbf{g}^i = \mathrm{GAP}\left(\boldsymbol{\mathcal{F}}_{s_4}^i\right) \in \mathbb{R}^{C_{s_4}}$. The multi-view representation and logits are

$$\mathbf{h} = \mathrm{Concat}\left(\mathbf{g}^1, \mathbf{g}^2, \dots, \mathbf{g}^N\right) \in \mathbb{R}^{NC_{s_4}}, \qquad \mathbf{z} = \mathbf{W}_c\,\mathbf{h} + \mathbf{b}_c \in \mathbb{R}^5, \tag{17}$$

The predictive distribution is

$$\hat{\mathbf{y}} = \mathrm{Softmax}(\mathbf{z}). \tag{18}$$

**Training objective:** The main grading loss uses the focal loss on the final prediction:

$$\mathcal{L}_{\mathrm{cls}} = \mathrm{Focal}(\hat{\mathbf{y}}, \mathbf{y}). \tag{19}$$

The overall training objective combines the main loss, the stage-wise auxiliary loss from GALP (Eq. 2), and the MoE load-balancing regularizer with nonnegative weights $\lambda_{\mathrm{aux}}$ and $\lambda_{\mathrm{load}}$:

$$\mathcal{L}_{\mathrm{total}} = \mathcal{L}_{\mathrm{cls}} + \lambda_{\mathrm{aux}}\,\mathcal{L}_{\mathrm{aux}} + \lambda_{\mathrm{load}}\,\mathcal{L}_{\mathrm{load}}. \tag{20}$$

## 4 EXPERIMENTS

### 4.1 EXPERIMENTAL SETUP

**Datasets:** We evaluate on two multi-view DR grading datasets used in prior work: MFIDDR (four-view) (Luo et al., 2023) and DRTiD (two-view) (Hou et al., 2022). MFIDDR contains 8,613 eyes, each with four fundus photographs captured from distinct angles. The provider also releases lesion segmentation masks generated by a segmentation model. The official split is 70/30 for training/testing. Following prior work on this benchmark, we resize each image to $224 \times 224$ for training and evaluation. In addition, following work (Hu et al., 2025), we preprocess the images in MFIDDR using code in Karthik et al. (2019). DRTiD comprises 3,100 eyes with two views per eye. The dataset is partitioned into 2,000 eyes for training and 1,100 for testing. For each image, the provider additionally supplies OD and macular coordinates. To enable fair comparison with the SOTA method CrossFiT (Hou et al., 2022), we resize images to $512 \times 512$.

Table 1: Performance comparison on the four-view MFIDDR dataset.

| Method (End-to-End) | Acc | Spe | Kappa | F1 |
|---|---|---|---|---|
| RETFound  (Zhou et al., 2023) | 74.1 | 73.8 | 48.4 | 70.9 |
| MVCINN  (Luo et al., 2023) | 80.1 | 83.3 | 62.5 | 78.9 |
| MVCNN_R  (Yu et al., 2020) | 77.4 | 79.2 | 56.6 | 79.2 |
| MVCNN_V  (Yu et al., 2020) | 79.1 | 80.5 | 59.9 | 77.2 |
| ETMC  (Han et al., 2022) | 81.5 | 83.4 | 64.8 | 79.7 |
| LFMVDR(w/o lesion)  (Luo et al., 2024) | 80.4 | 85.9 | 64.0 | 79.4 |
| **Method (Externally informed)** | | | | |
| CVSA (with vessel)  (Lin et al., 2025a) | 82.6 | 86.8 | 67.9 | 81.9 |
| WGLIN (with lesion)  (Hu et al., 2025) | 84.2 | 89.9 | 71.2 | 83.6 |
| SMVDR-W (with lesion)  (Luo et al., 2025) | 83.0 | 88.5 | 68.9 | 82.4 |
| SMVDR-M (with lesion)  (Luo et al., 2025) | 84.0 | **91.3** | 71.4 | 83.7 |
| LFMVDR (with lesion)  (Luo et al., 2024) | 82.2 | 86.9 | 66.9 | 81.3 |
| **Ours (w/o lesion)** | 83.9 | 89.8 | 70.9 | 83.5 |
| **Ours (with lesion)** | **84.6** | 90.6 | **72.3** | **84.4** |

**Implementation details.** Following prior fundus analysis work (Wang et al., 2024), we adopt Swin-Transformer (Swin-B) as the backbone. Following prior SOTA works on the two datasets, we initialize the backbone differently: for MFIDDR, the backbone is pretrained on ImageNet, consistent with CVSA (Lin et al., 2025a); for DRTiD, the backbone is pretrained on the fundus dataset EyePACS  (Dugas et al., 2015), following CrossFiT (Hou et al., 2022). To ensure that the patch size exactly divides the spatial dimensions of feature maps with different resolutions, we set the patch size to $q = 7$ for MFIDDR and $q = 8$ for DRTiD. Since the auxiliary loss $\mathcal{L}_{aux}$ is also a classification loss, we use $\lambda_{aux} = 1$, identical to the weight of $\mathcal{L}_{cls}$. The load-balancing weight is set to $\lambda_{load} = 0.1$. For all stages $s_n$, we retain $K_{1,s_n}$ tokens, corresponding to a retention ratio $\alpha = K_{1,s_n}/P_{s_n}$, where $P_{s_n}$ is the total number of tokens at stage $s_n$. In our experiments we fix $r = 50\%$. The expert pool contains $M = 6$ experts, with $K_2 = 2$ experts activated per routing step. For fair comparison with methods that utilize additional information, we also report results on MFIDDR using lesion annotations: lesion segments are fused with the original images via Spatially-Adaptive Denormalization (SPADE) (Park et al., 2019).

Table 2: Grade-wise Performance comparison on the four-view MFIDDR dataset.

| Method (End-to-End) | Grade 0 | | | Grade 1 | | | Grade 2 | | | Grade 3 | | | Grade 4 | | |
|---|---|---|---|---|---|---|---|---|---|---|---|---|---|---|---|
| | F1 | Pre | Spe | F1 | Pre | Spe | F1 | Pre | Spe | F1 | Pre | Spe | F1 | Pre | Spe |
| RETFound  (Zhou et al., 2023) | 87.5 | 80.1 | - | 35.9 | 50.2 | - | 49.4 | 54.4 | - | 66.7 | 65.8 | - | 36.7 | 90.0 | - |
| MVCINN  (Luo et al., 2023) | 91.3 | 86.7 | 75.9 | 56.4 | 68.3 | 94.1 | 59.3 | 57.4 | 95.8 | 68.1 | 70 | 97.9 | 44.8 | 68.4 | 99.7 |
| MVCNN_R  (Yu et al., 2020) | 89.4 | 83.6 | 69.3 | 46.1 | 64.1 | 94.1 | 59.4 | 58.1 | 95.9 | 68.4 | 66 | 97.3 | 22.2 | 83.3 | **99.9** |
| MVCNN_V  (Yu et al., 2020) | 90.1 | 84.5 | 71.1 | 50.0 | 65.3 | 94.3 | 60.2 | 65.3 | 94.3 | 73.6 | 66.8 | 97 | 38.5 | 76.9 | 99.8 |
| ETMC  (Han et al., 2022) | 91.8 | 86.8 | – | 63.7 | 73.3 | - | 55.4 | **66.4** | - | 70.2 | 64.4 | - | 0.9 | 0.1 | - |
| **Method (Externally informed)** | | | | | | | | | | | | | | | |
| CVSA  (Lin et al., 2025a) | 92.3 | 89.2 | 81.2 | 62.6 | **73.6** | **95.0** | 64.2 | 61 | 96 | 73.2 | 72.7 | **98.0** | **64.1** | 64.1 | 99.3 |
| WGLIN  (Hu et al., 2025) | **93.5** | 92.3 | 87.0 | 71.4 | 71.0 | 92.3 | 59.9 | 63.9 | **97.1** | 74.7 | 71.9 | 97.7 | 29.8 | 87.5 | **99.9** |
| SMVDR-W  (Luo et al., 2025) | 92.9 | 91.1 | - | 68.3 | 69.1 | - | 55.6 | 60.6 | - | 73.8 | 71.5 | - | 40.8 | **99.9** | - |
| SMVDR-M  (Luo et al., 2025) | **93.5** | **93.4** | - | **71.7** | 71.2 | - | 60.3 | 60 | - | 74.2 | 69.4 | - | 30.4 | **99.9** | - |
| LFMVDR  (Luo et al., 2024) | 92.4 | 89.7 | 82.1 | 66.3 | 69.5 | 92.7 | 59 | 62.1 | 96.8 | 70.9 | 69.5 | 97.6 | 17 | 50 | **99.9** |
| **Ours (w/o lesion)** | 93.4 | 92.0 | 86.5 | 69.7 | 72.2 | 93.2 | 62.5 | 62.6 | 96.5 | 74.1 | 70.3 | 97.6 | 36.0 | 81.8 | **99.9** |
| **Ours (with lesion)** | **93.5** | 92.7 | **87.9** | 71.4 | 72.3 | 93.0 | **65.2** | 65.4 | 96.8 | **74.8** | **73.4** | **98.0** | 51.6 | 69.6 | 99.8 |

**Metrics:** For evaluation on the four-view dataset, we follow the SOTA works (Luo et al., 2024; 2025) and report overall Accuracy (Acc), Specificity (Spe.), Cohen's Kappa (Kappa.), Precision (Pre) and F1 score. For the two-view dataset, we follow the protocol established in (Hou et al., 2022), which evaluates methods in terms of Accuracy and the Area Under the ROC Curve (AUC) for each DR grade.

Table 3: Performance comparison on the two-view DRTiD dataset.

| Method (End-to-End) | Acc | AUC | | | | |
|---|---|---|---|---|---|---|
| | | Grade 0 | Grade 1 | Grade 2 | Grade 3 | Grade 4 |
| Binocular Network (Qian et al., 2021) | 66.18 | 90.7 | 66.42 | 78.33 | 92.25 | 86.23 |
| Cv-Transformer (Van Tulder et al., 2021) | 69.45 | 92.2 | **73.6** | 81.9 | 94.6 | 95.7 |
| MVCNN_R (Yu et al., 2020) | 74.0 | 94.1 | 68.4 | 84.2 | 95.3 | 94.7 |
| MVCNN_V (Yu et al., 2020) | 73.8 | 93.7 | 67.4 | 84.4 | 94.8 | 93.3 |
| DeepDR (Dai et al., 2021) | 72.7 | 94.1 | 68.8 | 84.4 | 95.2 | 93.4 |
| **Method (Externally informed)** | | | | | | |
| CVSA (with vessel) (Lin et al., 2025a) | 74.7 | 93.8 | 66.4 | 84.1 | 95.7 | 95.5 |
| CrossFiT (with OD and macula) (Hou et al., 2022) | 75.6 | **94.7** | 70.2 | **85.8** | 95.5 | 95.4 |
| **Ours (End-to-End)** | **76.0** | 94.6 | 71.1 | 85.3 | **96.2** | **95.9** |

Table 4: Ablation study.

| Models | Acc | Spec | Kappa | F1 |
|---|---|---|---|---|
| w/o GALP | 82.7 | 88.5 | 68.5 | 82.1 |
| w/o Experts | 82.6 | 87.9 | 68.2 | 81.4 |
| w/o LGRF | 82.3 | 87.6 | 67.4 | 81.2 |
| **Ours (w/o lesion)** | **83.9** | **89.8** | **70.9** | **83.5** |

**Compared methods:** To evaluate the effectiveness of our proposed framework, we compare it against a comprehensive set of multi-view DR grading baselines. For clarity, the existing methods are grouped into two categories. **(1) End-to-end multi-view methods.** This group includes multi-view version of RETFound (Zhou et al., 2023), MVCINN (Luo et al., 2023), MVCNN_R (Resnet50 version of MVCNN) and MVCNN_V (vgg19 version of MVCNN) (Yu et al., 2020), Binocular Network (Qian et al., 2021), Cv-Transformer (Van Tulder et al., 2021), DeepDR (Dai et al., 2021) and ETMC (Han et al., 2022). These approaches operate in a purely end-to-end fashion, focusing on feature extraction and cross-view fusion without relying on additional annotations. **(2) Externally informed methods.** This group incorporates additional signals to enhance grading performance. Representative examples include CVSA (using vessel masks) (Lin et al., 2025a), WGLIN (wavelet-based lesion guidance) (Hu et al., 2025), SMVDR and LFMVDR (lesion maps) (Luo et al., 2024; 2025), and CrossFiT (OD and macula coordinates) (Hou et al., 2022). These methods demonstrate how auxiliary cues can improve DR grading, but at the cost of requiring additional annotations. Together, these two categories cover the mainstream directions of current research: purely end-to-end multi-view pipelines and annotation-augmented approaches. By comparing with both, we provide a fair and comprehensive evaluation of our method, highlighting its ability to retain the advantages of end-to-end training while reducing reliance on external annotations.

## 4.2 COMPARISON WITH SOTA METHODS

**Experiments on MFIDDR:** We first evaluate our method on the four-view MFIDDR dataset and compare it with a series of multi-view approaches. As shown in Table 1, our lesion-free variant achieves 83.9% accuracy, 89.8% specificity, 70.9% kappa, and 83.5% F1. This performance not only surpasses all end-to-end baselines, but also outperforms or matches several externally informed methods such as LFMVDR (with lesion) and CVSA (with vessel). This finding is noteworthy: even without any external side information, our framework already closes the gap with methods that require costly annotations, and in some cases performs better. The small residual differences with the strongest externally informed models are acceptable, given that our approach is fully external-annotation-free at inference. When lesion information is incorporated, our method further improves to 84.6% accuracy, 90.6% specificity, 72.3% kappa, and 84.4% F1, establishing new SOTA performance on this benchmark. This demonstrates that our architecture is inherently effective, and that externally provided lesion cues can be integrated to deliver further gains. As summarized in Table 2, our lesion-free model already surpasses several externally guided methods, showing strong Grade 0–3 performance, particularly in Grade 3 (F1=74.1%). With lesion input, our approach further im-

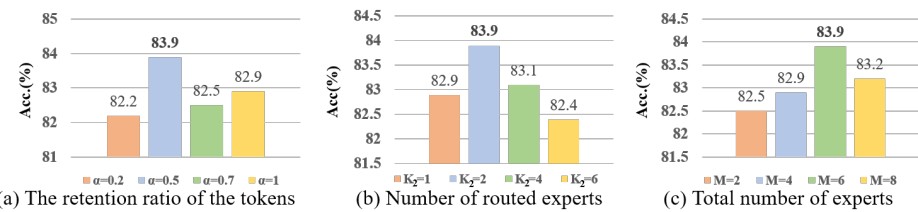

Figure 3: The hyperparameter analysis.

proves, reaching the best Grade 2 (F1=65.2%) and Grade 3 (F1=74.8%). Importantly, even without extra cues, performance is comparable to annotation-based methods, while with lesions we achieve SOTA.

**Experiments on DRTiD:**  As shown in Table 3, our end-to-end approach achieves the highest overall accuracy, outperforming all existing methods, including all externally informed methods (CVSA and CrossFiT), even though our method does not require any external annotations. In terms of AUC, our method consistently achieves competitive or superior results across different grades (best on grade 3 and 4).

In summary, our method demonstrates two key advantages. First, even without external annotations, it achieves or surpasses the performance of most of externally informed approaches, showing that lesion proposals generated internally by GALP are effective surrogates for expert cues. Second, when external information is available, our framework can incorporate it and attain state-of-the-art results. These results confirm the robustness and adaptability of our proposed architecture.

### 4.3 Ablation and hyperparameter study

We conducted ablation experiments on the MFIDDR dataset to evaluate the contributions of the proposed modules. As shown in Table 4, w/o GALP removes the GALP mechanism and directly uses all tokens for LGRF fusion; w/o Experts discards the expert pool and directly applies cross-attention on lesion proposals; w/o LGRF eliminates the fusion module and simply concatenates lesion proposals with cross-view tokens. In addition, we conduct a hyperparameter study on $K_{1,s_n}$ and $K_2$ (see Fig. 3). For $K_{1,s_n}$ at each stage $s_n$, we vary the retention ratio $\alpha \in \{0.20, 0.50, 0.70, 1.00\}$ of tokens kept as lesion proposals. For $K_2$ (the number of activated experts), we test values $\{1, 2, 4, 6\}$. For $M$ (the total number of experts), we test values $\{2, 4, 6, 8\}$ As shown in Table 4, the ablation study demonstrates that both GALP and LGRF play crucial roles in enhancing performance: removing either module leads to clear drops in accuracy, kappa, and F1, confirming their complementary benefits; eliminating the expert pool further weakens fusion effectiveness. From Fig. 3, the hyperparameter analysis shows that retaining 50% of tokens ($\alpha = K_{1,s_n}/P_{s_n}$) yields the best trade-off between accuracy and redundancy, while activating $K_2 = 2$ experts in the expert pool of $M = 6$ provides the most stable and accurate results, balancing diversity and computational efficiency.

## 5 Conclusion

In this work, we propose a novel end-to-end framework for multi-view DR grading that integrates lesion-aware cues without requiring external annotations. The proposed GALP module strengthens stage-wise feature discriminability through auxiliary classification and transforms grade-conditioned evidence maps into lesion proposals, which act as surrogates for costly expert cues. The LGRF module further enables context-aware cross-view fusion by dynamically routing experts and applying Top-$K$ weighted cross-view attention, ensuring precise and selective integration of lesion proposals across views. Extensive experiments on two multi-view fundus datasets, MFIDDR and DRTiD, demonstrate that our method achieves SOTA performance. Importantly, the proposed framework attains accuracy comparable to models that rely on external annotations, suggesting its practical potential for clinical deployment where such additional data are often unavailable or costly to obtain. By reducing annotation dependency while maintaining high diagnostic accuracy, our method provides a label-efficient and scalable solution for large-scale DR screening.

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

# A  APPENDIX

## A.1  USE OF LARGE LANGUAGE MODELS (LLMs)

We used large language model solely as a writing assistant to improve the clarity, grammar, and style of the manuscript. The model was not involved in research ideation, experimental design, implementation, analysis, or result interpretation. We sincerely appreciate the contribution of the large language model in enhancing the readability and linguistic quality of this work. Its assistance was instrumental in refining the presentation of our research. All technical content, including methods, experiments, and conclusions, was fully developed and verified by the authors. The authors take full responsibility for the content of this paper.

Table 5: Comparison results against single-view approaches with our proposed multi-view approach.

| Method | Acc | Spe | Kappa | F1 |
|---|---|---|---|---|
| Inception_ResNet_k_2 (Szegedy et al., 2017) | 70.6 | 67.1 | 38.6 | 65.4 |
| Mobile_Net_k2 (Sandler et al., 2018) | 72.3 | 68.7 | 43.6 | 67.2 |
| ResNet50 (He et al., 2016) | 73.2 | 73.2 | 45.2 | 69.3 |
| ResNext50_32x4d (Xie et al., 2017) | 73.3 | 73.0 | 47.1 | 70.3 |
| ConvNeXt-B (Liu et al., 2022) | 75.9 | 77.8 | 53.7 | 73.6 |
| Swin-B (Liu et al., 2021) | 75.0 | 75.5 | 51.3 | 72.4 |
| Vim (Zhu et al., 2024) | 77.0 | 81.2 | 56.3 | 75.3 |
| PVT-M (Wang et al., 2021) | 74.1 | 78.5 | 50.4 | 71.4 |
| PVT-L (Wang et al., 2021) | 75.3 | 80.3 | 57.2 | 73.8 |
| RETFound (Zhou et al., 2023) | 71.7 | 70.9 | 43.6 | 67.3 |
| **Ours** | **83.9** | **89.8** | **70.9** | **83.5** |

## A.2  COMPARISON WITH SINGLE-VIEW METHODS

We compare our approach with single-view methods on MFIDDR in Table 5. Our model is trained on all four views jointly, whereas each single-view method is trained separately on each view and

the best-performing view is reported for comparison. As shown in Table 5, our method attains the highest Accuracy, Specificity, Kappa, and F1. These results underscore the benefit of multi-view learning: aggregating complementary cross-view information yields clear gains over the strongest single-view baselines.

### A.3 PSEUDO TRAINING CODE

The training process of our method can be seen in Algorithm 1.

---

Algorithm 1: The training process of our method

---

**Input**: Multi-view Fundus images $\boldsymbol{\mathcal{V}} = \left[\boldsymbol{\mathcal{V}}^1, \boldsymbol{\mathcal{V}}^2, \ldots, \boldsymbol{\mathcal{V}}^N\right]$ and its corresponding grading label $\boldsymbol{y}$.

**Parameters**: The retention ratio $\alpha$ of the tokens in the GALP. The number of routed experts $K_2$. The total number of experts $M$.

**Output**: A trained model.

1: **for** $ep$=1 to Epo **do**
2:     Pre-process the images
3.     Extract initial features from the Conv Block
4.     **for** $n$=1 to 3 **do**
5.         Extract encoded features $\boldsymbol{\mathcal{F}}_{s_n}^i$ from the Encoder in the stage-$s_n$
6.         Compute the auxiliary loss and the lesion proposals $\boldsymbol{\mathcal{P}}_{s_n}^i$ with the ratio $\alpha$
7.         Compute the routing scores $\boldsymbol{\mathcal{R}s}_{s_n}^i$ and extract the Top-$K_2$ experts outputs $\boldsymbol{\mathcal{P}e}_{s_n,k_2}^j$
8.         Compute the MoE load-balancing and the cross-view fused results $\mathbf{Fu}_{s_n}^i$
9.     **end for**
10.    Extract features from $\mathbf{Fu}_{s_3}^i$ by the Encoder in stage $s_4$, obtaining $\boldsymbol{\mathcal{F}}_{s_4}^i$
11.    Predict the grade on the $\boldsymbol{\mathcal{F}}_{s_4}^i$ and compute the grading loss
12:    Update gradient.
13: **end for**

---

