# OpenReview forum: "Expert-Guided Cross-View Fusion with Self-Derived Lesion Proposals for Multi-View Diabetic Retinopathy Grading"
_ICLR.cc/2026/Conference — ICLR 2026 Conference Withdrawn Submission_

### Official Review · Reviewer_WVta · 2025-10-28

**Soundness:** 3
**Presentation:** 3
**Contribution:** 3
**Rating:** 6
**Confidence:** 3

**Summary:**

This paper presents an end-to-end framework for multi-view DR grading. It directly confronts a major challenge in automated DR detection: standard deep learning models (e.g., CNNs & Transformers) often "lose" small, low-contrast lesions (e.g., microaneurysms) due to downsampling and tokenization. While prior work has compensated for this by using costly, manually-annotated external data (e.g., lesion maps, vessel segmentations), this paper proposes a method that achieves similar or better performance without relying on these expensive annotations. It does this by "self-deriving" its own lesion proposals on the fly and using a cross-view fusion mechanism to focus on them. Empirical performance on  four-view MFIDDR and two-view DRTiD datasets show promising performance.

**Strengths:**

- The proposed Grade-Activated Lesion Proposal (GALP) module is interesting and sound, which drives the self-generated lesion proposal to capture small and low-contrast lesions.

- The proposed Cross-view lesion proposal expert routing adaptively and precisely fuses important information from multiple fundus images.

- The empirical performance of the proposed method is promising. Ablations on different module components are provided, which demonstrates the effectiveness of different components.

- The paper is, in general, well-structured.

**Weaknesses:**

- It seems to me that the improvements are quite marginal, and sometimes performance of the proposed method is tie to other baselines. It is quite hard to judge if the improvement is statistically meaningful or just by chance (as no cross-validation is conducted).

- The proposed framework, with its GALP module and LGRF modules (which uses a Mixture-of-Experts, routing, and weighted attention), seems more complex than those baselines. This added complexity could lead to higher computational costs for both training and inference. The additional computational overhead should be discussed and acknowledged in the paper.

- The ablation study (Figure 3) shows that the model's performance is quite sensitive to its specific sets of hyperparameters. For instance, its top performance was achieved with a token retention ratio of 50% (a=0.5), 2 routed experts (K2=2), and a total of 6 experts (M=6). Performance largely drops off with other values. Would be optimal hyperparameters for different datasets be largely different ? If so, It seems challenging in practice to determine these hyperparameters for different datasets.

**Questions:**

- Can the authors provide some visualizations of the attention map to confirm that the small lesions are indeed captured by the proposed method ? Or, ideally, quantitative results on the localization of lesions should be provided to better support the main claim.

- Can the authors also provide comparison on Eyepacs for the single-view experiment, and compare with more state-of-the-art methods for DR grading, e.g., [1].

[1] Lesion-Aware Transformers for Diabetic Retinopathy Grading.

---

### Official Review · Reviewer_33z8 · 2025-10-31

**Soundness:** 3
**Presentation:** 2
**Contribution:** 2
**Rating:** 4
**Confidence:** 4

**Summary:**

The paper proposes an end-to-end framework for multi-view DR grading that avoids costly external annotations by (1) generating self-derived, grade-conditioned lesion proposals via a GALP module, and (2) performing selective cross-view fusion with LGRF module using gated mixture-of-experts and Top-K–weighted cross-view attention. Evaluated on 2 datasets, the approach matches or exceeds strong end-to-end baselines and is competitive with “externally informed” methods that use vessel/lesion/OD cues.

**Strengths:**

- The paper is well-motivated and has reasonable designs.

- Clear modular design: GALP (auxiliary, grade-conditioned CAMs → Top-K proposal tokens) and LGRF (MoE-gated, Top-K cross-view attention) are described with implementable details.

- The authors provide a comprehensive ablation study in Table 4 and Figure 3. They not only compare the effectiveness of modules, but also that of the MoE’s hyperparameters.

**Weaknesses:**

- The paper’s core claim is that the GALP module generates effective ‘lesion proposals’ that act as ‘surrogates for external cues’, and Figure 1 shows that this model has better interpretability. However, the paper provides no qualitative visualizations or the resulting Top-K proposals to support designs and assumptions.

- The proposed method adds considerable complexity over a standard Swin-Transformer backbone: three auxiliary classifiers , CAM generation, Top-K selection , and a multi-stage MoE-based fusion module (LGRF). An analysis of the added computational overhead (e.g., FLOPs, parameters, and inference speed) would be helpful.

- Results are on two datasets from the same modality; there is no cross-dataset generalization experiments.

**Questions:**

Please see Weakness part.

---

### Official Review · Reviewer_RZF8 · 2025-10-31

**Soundness:** 3
**Presentation:** 2
**Contribution:** 2
**Rating:** 4
**Confidence:** 4

**Summary:**

In this manuscript, the authors propose a diabetic retinopathy grading method with multi-view fundus images. Two modules, GALP and LGRF, are designed to generate lesion proposals and provide self-guidance for grading. The GALP module utilizes an auxiliary classification head and selects the top k regions as proposals, which are used to guide the fusion across views in the LGRF module. Experimental results prove the effectiveness of the proposed method.

**Strengths:**

1. The proposed method demonstrates its stable and superior performance by comparing with a wide range of recent SOTA methods.
2. Detailed ablation studies are conducted to select optimal hyperparameters.
3. Well-established literature review for the audience to quickly grasp key content in this field.

**Weaknesses:**

1. **Limited Novelty:** The claimed innovation appears to be a repackaging of established concepts under a new nomenclature. The proposed Grade-Activated Lesion Proposal (GALP) module consists of two largely independent components: a CNN+MLP network for grade prediction (trained with an auxiliary loss) and a mechanism that utilizes an attention map to select Top-K patches, which are then encoded as tokens for subsequent modules. The core concept of "grade-activated" is unclear, as the final prediction target is the grade itself; the network would arguably be grade-informed even without the auxiliary loss. Furthermore, the term "lesion proposal" is questionable, as there is no indication that explicit lesion annotations are utilized anywhere in the pipeline to guide the information flow. What is termed a "grade-conditioned evidence map" functionally resembles a standard attention map.
2. **Unnecessarily Complex Mathematical Notation:** The mathematical presentation is often overly convoluted, which hinders readability. For instance, the GALP and LGRF modules are functionally identical across different stages. A unified description would suffice, eliminating the need for stage-specific subscripts like $s_n$ in terms such as $\mathcal{I}^i_{s_n}$ and $\mathcal{F}^i_{s_n}$. Moreover, once a set $\mathcal{I}^i_{s_n}$ is defined, its elements can be represented by a simple variable (e.g., $j$) without the need for the complex notation $k_{1,s_n} \in \mathcal{I}_{s_n}^i$. Streamlining this notation would significantly improve the clarity of the manuscript.
3. **Insufficient Implementation Details:** The description of the methodology lacks critical details, which hinders reproducibility. Specific omissions will be outlined in the 'Questions' section.

**Questions:**

1. In Eq. 1, how is the variable $W_{s_n}$ calculated? No definition is provided.
2. In Eq. 3, how is the weight vector $w_{s_n}^{(\hat{y}_{s_n}^i)}$ calculated? Or is it a trainable variable? And why not use $i$ as the superscript?
3.  In Fig. 2, it seems that all features $\mathcal{F}^i_{s_n}$ are fused into one evidence map in GALP. But according to the text, each view has a unique map. Which is correct?
4. How is the number of stages decided?
5. In performance comparisons, the proposed method has two versions, with and w/o lesion. What are the differences in the implementation?
6. Are you sure that CrossFit utilized the locations of the optic disc and macula as external information? I think it is also an end-to-end method.

---

### Official Review · Reviewer_zGcY · 2025-11-01

**Soundness:** 3
**Presentation:** 4
**Contribution:** 2
**Rating:** 4
**Confidence:** 4

**Summary:**

This paper proposes an end-to-end multi-view diabetic retinopathy (DR) grading framework that integrates two key modules: GALP generates self-derived lesion proposals using auxiliary classifiers and class activation maps (CAMs). And LGRF, which fuses features across different fundus views using a gated MoE mechanism and Top-K cross-view attention. Experiments are conducted on two public multi-view DR datasets (MFIDDR and DRTiD), and the proposed model reports slightly higher accuracy than several existing end-to-end and annotation-guided baselines.

**Strengths:**

(1). Clear Motivation – The paper targets the practical issue of reducing dependence on expensive manual lesion annotations for DR grading.

(2). Complete Pipeline – The method combines multi-stage auxiliary supervision and cross-view fusion in a unified end-to-end structure.

(3). Good Presentation – Writing is clear, figures are informative, and mathematical details are mostly complete.

(4). Comprehensive Experiments – Results are reported on multiple datasets with ablation and hyperparameter analyses.

**Weaknesses:**

(1). Limited novelty and reliance on existing techniques

The GALP module is essentially a standard CAM-based lesion proposal mechanism combined with an auxiliary classifier, rather than a genuinely novel component. It closely resembles a straightforward extension of the Class Activation Map (CAM) concept [1].

The LGRF module’s MoE gating and cross-view fusion largely reuse existing mixture-of-experts or cross-attention architectures without substantial methodological innovation, following similar ideas to Cao et al. (2023) and Xie et al. (2025) [2,3].


[1] Peng-Tao Jiang, Chang-Bin Zhang, Qibin Hou, Ming-Ming Cheng, and Yunchao Wei. LayerCAM: Exploring hierarchical class activation maps for localization. IEEE Transactions on Image Processing, 30:5875–5888, 2021.

[2] Bing Cao, Yiming Sun, Pengfei Zhu, and Qinghua Hu. Multi-modal gated mixture of local-to-global experts for dynamic image fusion. In Proceedings of the IEEE/CVF International Conference on Computer Vision (ICCV), pp. 23555–23564, 2023.

[3] Luyuan Xie, Tianyu Luan, Wenyuan Cai, Guochen Yan, Zhaoyu Chen, Nan Xi, Yuejian Fang, Qingni Shen, Zhonghai Wu, and Junsong Yuan. DFLMoE: Decentralized federated learning via mixture of experts for medical data analysis. In Proceedings of the IEEE/CVF Conference on Computer Vision and Pattern Recognition (CVPR), pp. 10203–10213, 2025.

(2). Lack of theoretical insight and marginal performance gains

Although the method achieves slightly higher accuracy (≈ +0.4–0.6 %) than some prior works, such as LFMVDR and CVSA, the improvement is within experimental noise, and no statistical significance or confidence interval is provided. The paper does not convincingly explain why self-generated lesion proposals perform comparably to expert annotations.This design suffers from a chicken-and-egg problem: lesion proposals depend on the model’s current grading predictions, yet those predictions are later refined using the same proposals. Such mutual dependency means that good localization implies good prediction, but if the early training is unstable, poor proposals may reinforce incorrect cues.
Unlike other weakly supervised methods that employ progressive or curriculum learning to address this issue [1], the authors provide no mechanism to stabilize.

[1] Yunpeng Chen et al. Progressive Deep Supervision for Weakly Supervised Object Localization. ICCV, 2017.

(3).  Missing visualization and validation of GALP

Although GALP is central to the contribution, no explicit visualization or analytical validation is provided. The authors claim that GALP can automatically generate lesion-aware proposals, but they do not show any heatmaps, qualitative lesion maps, or quantitative comparisons to expert-annotated regions.

**Questions:**

See weakness above

---

### Note · Authors · 2025-11-26

I have read and agree with the venue's withdrawal policy on behalf of myself and my co-authors.